# Enabling 3D CT-scanning of cultural heritage objects using only in-house 2D X-ray equipment in museums

Francien G. Bossema [1,2] ✉, Willem Jan Palenstijn [1,3], Arlen Heginbotham [4], Madeline Corona [4], Tristan van Leeuwen [1,5], Robert van Liere [1,6], Jan Dorscheid [2], Daniel O'Flynn [7], Joanne Dyer [7], Erma Hermens[8] & K. Joost Batenburg[1,3]

Visualizing the internal structure of museum objects is a crucial step in acquiring knowledge about the origin, state, and composition of cultural heritage artifacts. Among the most powerful techniques for exposing the interior of museum objects is computed tomography (CT), a technique that computationally forms a 3D image using hundreds of radiographs acquired in a full circular range. However, the lack of affordable and versatile CT equipment in museums, combined with the challenge of transporting precious collection objects, currently keeps this technique out of reach for most cultural heritage applications. We propose an approach for creating accurate CT reconstructions using only standard 2D radiography equipment already available in most larger museums. Specifically, we demonstrate that a combination of basic X-ray imaging equipment, a tailored marker-based image acquisition protocol, and sophisticated data-processing algorithms, can achieve 3D imaging of collection objects without the need for a costly CT imaging system. We implemented this approach in the British Museum (London), the J. Paul Getty Museum (Los Angeles), and the Rijksmuseum (Amsterdam). Our work paves the way for broad facilitation and adoption of CT technology across museums worldwide.

The interior of an art object often contains answers to questions about how and when the object was made, where the materials came from, and in some cases even who made it. This information can potentially be revealed by computed tomography (CT), a powerful technique for creating a three-dimensional (3D) image of the interior of an object. CT imaging was originally developed for health care[1], but also has applications in industry[2] and cultural heritage[3]. Cultural heritage research has used CT imaging of artifacts to determine their manufacturing process[4], current state[5–7], and origin[8]. Over the past years, the possibilities of CT imaging have been expanded by applying image processing methods to CT data, for example when unfolding unopened documents[9], combining CT data with other 3D imaging methods[10], and applying deep learning techniques to improve resolution[11].

To indirectly observe internal features, most museums have resorted to 2D X-ray imaging equipment, which can be straightforwardly applied to objects of various sizes and shapes. Typically, this equipment is used in a radiation-shielded room, which provides

[1]Centrum Wiskunde & Informatica, Amsterdam, The Netherlands. [2]Rijksmuseum, Amsterdam, The Netherlands. [3]Leiden Institute of Advanced Computer Science, Universiteit Leiden, Leiden, The Netherlands. [4]The J. Paul Getty Museum, Los Angeles, USA. [5]Universiteit Utrecht, Utrecht, The Netherlands. [6]Technische Universiteit Eindhoven, Eindhoven, The Netherlands. [7]British Museum, London, UK. [8]Fitzwilliam Museum, Cambridge University, Cambridge, UK. ✉e-mail: bossema@cwi.nl

extensive flexibility for imaging of large and irregularly shaped objects that do not fit in the confines of medical or cabinet-based CT scanners. In 2D radiography, the internal features of an object are projected onto a single image, which results in the loss of depth information. A 3D CT reconstruction volume, on the other hand, can be sliced to investigate interior features of the object at their exact 3D location within the object. Performing a CT scan requires a dedicated CT scanner, which acquires a sequence of 2D radiographs from angles all around the object and records the geometrical parameters needed for the mathematical reconstruction algorithm, which computes a 3D image of the object's interior. To facilitate this rotational acquisition, either the X-ray source and detector are mounted on a gantry that rotates around the static object, or the object is placed on a turntable that moves with respect to a static X-ray source and detector. In both cases, the stability and accuracy of all components are dependent on sophisticated system design combined with high-quality computer-controlled stages, as well as extensive system calibration to precisely control the orientation and timing of each radiograph.

Despite the capabilities of CT imaging, its use in cultural heritage research is still limited to selected cases, often carried out offsite. For example, clinical CT has been carried out on paintings[12] and mummies[13]. Since costly commercial-class micro-CT systems often provide higher resolution images, these have been used for purposes such as dendrochronology[8,14], analyzing panel paintings[15,16], and investigating unopened letters[9]. These systems are focused on a specific object dimension range, due to the detector size and the space within the cabinet, which limits their versatility for the broad range of object sizes and shapes in museum collections. An even less accessible option is synchrotron facilities, which can provide high-resolution images of small objects[17].

Although CT scanning provides considerably more information than radiography, there are challenges specific to its use on cultural heritage objects, which are unique, precious, and often fragile. Moving objects to a scanning facility can be costly because of specialized transport and insurance. Another challenge is the objects' wide variety

of sizes, shapes, and materials, which means the acquisition has to be tailored to the object[18,19]. Museums have addressed these challenges with a variety of setups. One example is a portable CT imaging setup which can be moved to investigate the object in situ[20–22]. Other solutions were sought out by the J. Paul Getty Museum (Los Angeles) which built a custom acquisition setup to investigate a bronze statue[23], and the British Museum (London), which obtained an easily accessible but costly in-house CT scanning facility[4].

In this article, we present an alternative approach for creating 3D CT imaging capabilities that can be applied to any existing radiography setup. By using a combination of basic X-ray imaging equipment, a tailored marker-based image acquisition protocol, and sophisticated data-processing algorithms, we can achieve 3D imaging of collection objects, alleviating the need for a costly CT system and making optimal use of the hardware already available. We demonstrate the efficacy of our approach by performing CT scans using the available X-ray imaging equipment at the British Museum, London; the J. Paul Getty Museum, Los Angeles; and the Rijksmuseum, Amsterdam. We imaged a small wooden block as a test object in all three museum radiography suites as well as in the FleX-ray lab micro-CT facility, situated at the Center for Mathematics and Computer Science in Amsterdam. We compared the results of our algorithms with those obtained using the well-calibrated in-house CT system already in use at the British Museum and the micro-CT system at the FleX-ray lab. The capacities of this technique and the new research possibilities it provides are further demonstrated by imaging a case study object at the J. Paul Getty Museum: a 19th-century plaster model *Python Killing a Gnu* by French artist Antoine-Louis Barye (1796–1875). Our approach enables 3D CT imaging, to the best of our knowledge for the first time, at the Getty Museum and Rijksmuseum radiography suites.

## Results

Our approach for creating accurate CT reconstructions uses only basic 2D radiography equipment and does not require precision operation of the moving parts, but instead relies on a set of markers (small metal balls) that are used to track all geometrical system parameters during image acquisition. This enables us to computationally derive the geometric system parameters that are typically hardware-calibrated in standard CT systems[24]. The radiographs acquired and system parameters calculated are combined to obtain a 3D CT reconstruction, which can be inspected to gain information about the interior features of the object.

### CT workflow

Our complete workflow for computing a 3D CT reconstruction from a series of standard 2D radiography measurements is illustrated in Fig. 1. The work carried out in the X-ray suite starts by placing small metal balls or *markers* in a piece of foam that surrounds the object. The object and marker holder are then placed on the rotation stage. The next step is the acquisition of radiographs in a full circular range, which yields a dataset with the markers in view. The computational workflow performed afterward consists of the following steps: (1) marker detection and labeling; (2) system parameter derivation; (3) preprocessing (flat- and dark-field correction) and removing the markers by inpainting; and (4) 3D reconstruction. The outputs corresponding to each of these steps are (1) labeled marker trajectories, (2) accurate system parameters, (3) preprocessed radiographs with the markers removed, and (4) 3D reconstruction based on the two previous outputs. For details on the methods and implementation, please refer to Section 4 and the Supplementary Methods.

### Comparison of one object imaged at three museum radiography suites and a micro-CT facility

We applied our methods to radiography datasets recorded at the research facilities of three prestigious museums: the British Museum

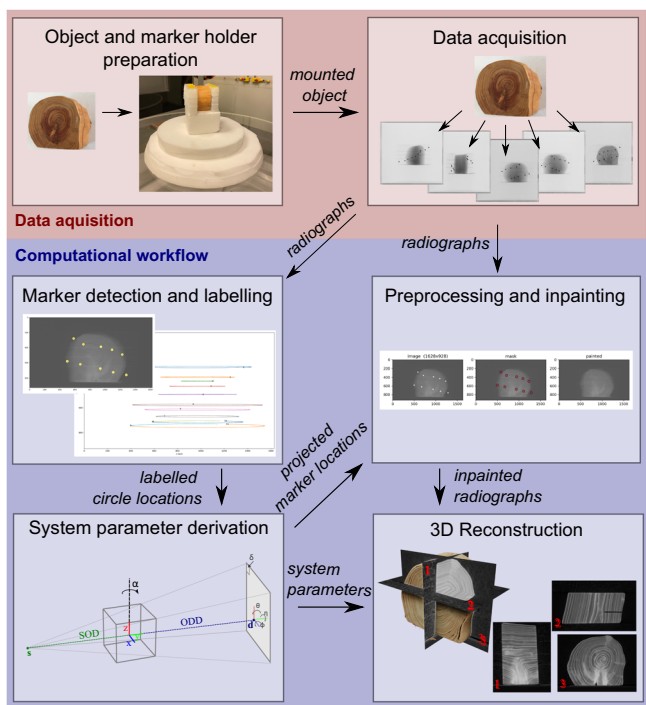

**Fig. 1** | Steps in the workflow for post-scan marker-based parameter derivation method for 3D reconstruction.

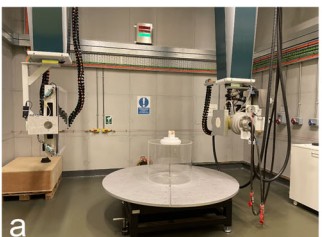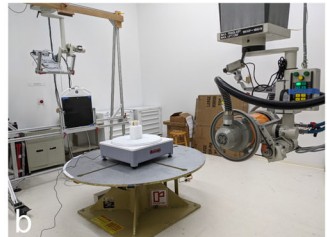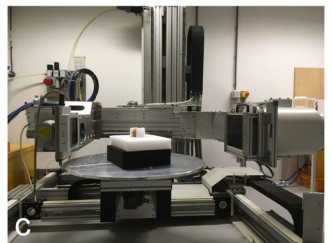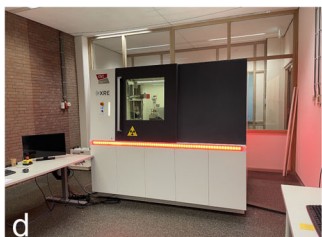

**Fig. 2 | X-ray imaging facilities. a** The British Museum (London), **b** the J. Paul Getty Museum (Los Angeles), **c** the Rijksmuseum (Amsterdam), and **d** the FleX-ray laboratory (Amsterdam).

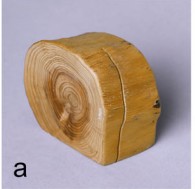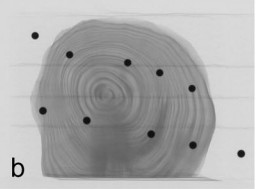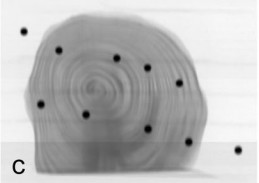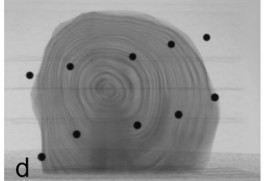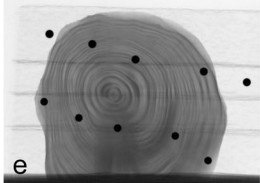

**Fig. 3 | The wooden test object. a** Wooden object (h 5 cm x w 6 cm x d 3 cm). Zoomed radiographs of the wooden test object at **b** the British Museum, **c** the J. Paul Getty Museum, **d** the Rijksmuseum, and **e** the FleX-ray laboratory.

(London)[25], the J. Paul Getty Museum (Los Angeles)[26], the Rijksmuseum (Amsterdam)[27], and a micro-CT facility: the FleX-ray laboratory (Amsterdam)[28], see Fig. 2. A small wooden object (h 5 cm x w 6 cm x d 3 cm, Fig. 3a), was scanned at all four facilities. The woodblock used for reconstructions was microscopically identified as yew (*Taxus* spp.)[29]. The details on the scanning parameters can be found in Table 1. In Fig. 3b−d we show sample radiographs of the object from each facility and in Fig. 4 five cross-sections of the CT reconstruction of the wooden block obtained from (1) the British Museum setup with system reported parameters, (2) the British Museum setup with post-scan marker-based parameter retrieval, (3) the J. Paul Getty Museum setup with post-scan marker-based parameter retrieval, (4) the Rijksmuseum setup with post-scan marker-based parameter retrieval and (5) the FleX-ray CT system with system reported parameters. As expected, the CT reconstructions obtained from the museum facilities using post-scan marker-based parameter retrieval do not reach the same effective resolution that can be observed in the micro-CT. The images show that the reconstruction with markers reveals the same internal structures as the reference reconstruction from the British Museum, which is based on their usual workflow for CT reconstructions.

There are several factors that potentially influence the image quality, such as the focal spot size of the source and the distances between the source, object, and detector. For example, the effect of the larger focal spot size in the J. Paul Getty Museum setup is visible on the radiograph (Fig. 3c), on which the markers are more blurred than in the other two facilities. Here, the source-to-detector distance and object-to-detector distance were chosen to match the British Museum distances for comparability, but these could be determined differently to improve the acquisition. We find that the angular increment is constant at the British Museum facility, but shows a more step-like profile in the other two facilities. Please see the Supplementary Figs. 3 and 4. The interior features of interest are shown in all reconstructed 3D images: the tree rings in the wood and the saw cut. The line profiles show that the contrast is sufficient to distinguish the tree rings. We observe that the image quality and detail in the line profile are considerably higher in the British Museum setup, whose system was intended for CT imaging. As we were working with uncalibrated systems in which multiple hardware and software factors may play a role, we could not exactly pinpoint the reasons for the differences in image quality.

Compared to the radiographs (Fig. 3), where the internal features are superimposed, the added advantage of the CT image is evident, since we gain depth information about the internal features and can slice the object open digitally. These CT slices allow further analysis of internal features.

Although the relatively low CT image quality at the J. Paul Getty Museum and Rijksmuseum facilities limits the use of automated post-processing tools to extract quantitative metrics from the data, the marker-based 3D reconstruction makes it possible to obtain digital cross-sections of objects and is highly useful for visual inspection of the interior features of objects. For cultural heritage objects, this implies a considerable knowledge gain with respect to radiographs. This will be further illustrated in the next section, with a case study scanned at the J. Paul Getty Museum. Notably, to the best of our knowledge, this is the first time the basic in-house radiography setups at the J. Paul Getty Museum and Rijksmuseum have been used for 3D X-ray CT reconstruction.

## Case study at the J. Paul Getty Museum

To further test the capabilities of the 3D CT reconstruction method and its application to the investigation of museum objects, we chose a case study from the J. Paul Getty Museum's collection: the plaster model *Python Killing a Gnu* by Antoine-Louis Barye (1796–1875) (Fig. 5a, collection number 85.SE.48), h 27.9 cm, w 39.1 cm, d 20.5 cm, here referred to as *the Barye model*. The Barye model is a complex construction consisting of plaster, metal armature, modeling wax,

### Table 1 | Scan settings of the small wooden block

| Scan settings | BM | GM | RM | FleX-ray |
|---|---|---|---|---|
| Tube voltage (kV) | 60 | 60 | 65 | 60 |
| Tube current (mA) | 3 | 15 | 4.5 | 800 |
| Focal spot size (mm) | 0.4 | 2.5 | 0.5 | 0.17 |
| number of projections | 1800 | 1469 | 1350 | 1440 |
| number of rounds | 1 | 3.05 | 2 | 1 |
| exposure time (s) | 1.0 | 0.9 | 0.1 | 0.4 |
| total scanning time (min) | 99 | 295 | 3.5 | 13 |

Scan settings as used at the British Museum (BM), the J. Paul Getty Museum (GM), the Rijksmuseum (RM), and the FleX-ray laboratory.

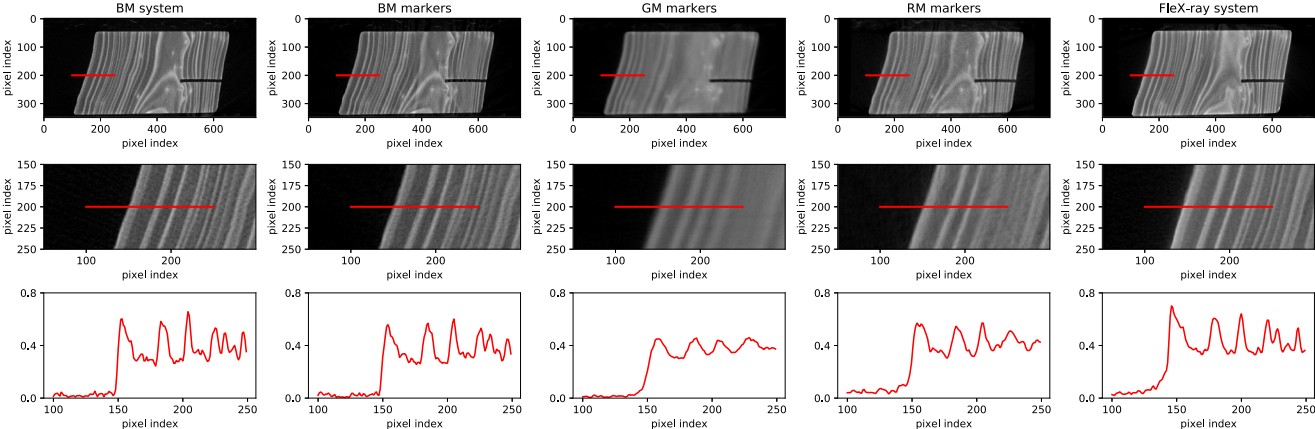

**Fig. 4 | Scan results of the wooden test object.** Top row. Single horizontal CT slice from reconstructions using the British Museum standard reconstruction workflow based on system feedback (BM system), all three museum systems with marker-based parameter retrieval (BM markers, GM markers, and RM markers), and the FleX-ray setup with system feedback. After reconstruction, the resulting 3D volumes have been scaled and registered in order to show a similar slice through the object using the FleXbox toolbox[49]. The intensities were normalized. The red line measures 1.5 cm. The tree rings and the saw cut are visible in all reconstructions. Middle row. Zoomed-in CT slices. Bottom row. A line profile of the normalized intensities corresponding to the red line in the reconstruction.

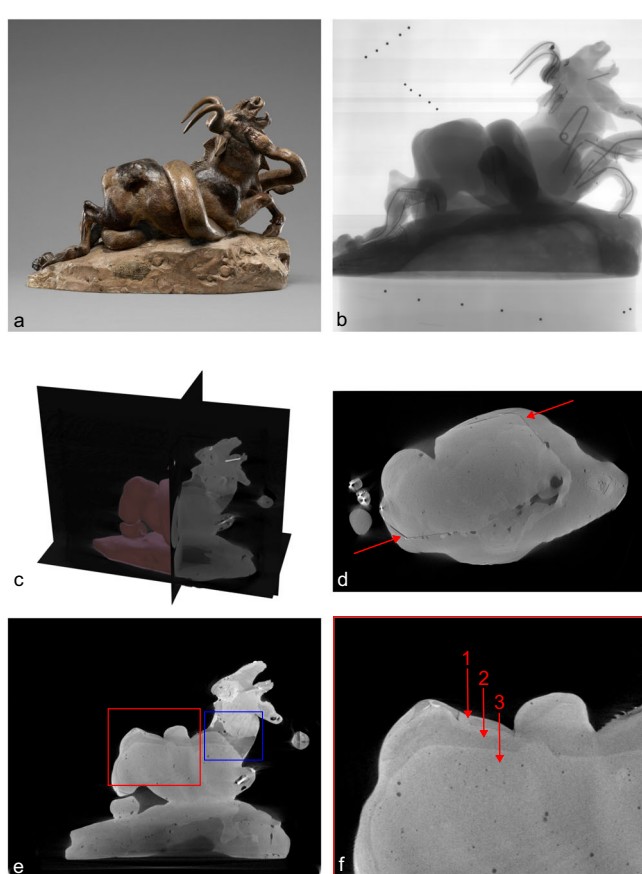

**Fig. 5 | Results of scanning the case study. a** The sculpture *Python Killing a Gnu* (1840s–1860s), Antoine-Louis Barye (French, 1796 –1875), the J. Paul Getty Museum collection number 85.SE.48, h 27.9 cm, w 39.1 cm, d 20.5 cm. **b** Single radiograph of the Barye model *Python Killing a Gnu*, including the markers used to determine the system parameters for CT reconstruction. **c** Three orthogonal slices of the CT reconstruction. **d** Horizontal slice, red arrows indicating the lines that show where the original square base is contained in the sculpture. **e** Vertical slice, the blue box indicates where gaps in the reconfigured neck were filled with wax instead of plaster. **f** Enlargement of the red box in **e**, the arrows indicate the three different layers of plaster used to create the sculpture.

paint, and adhesive, and contains numerous repairs executed in unknown materials. An ongoing technical study of this sculpture focuses on its complex history of use and its relationship with several related Barye bronzes in other collections.

The earliest Barye sculpture of a python killing a gnu (or wildebeest) is a bronze that was part of one of the artist's earliest and largest commissions: a *surtout de table* (centerpiece) commissioned by Ferdinand Philippe, duc d'Orleans, in 1834[30]. This bronze (Walters Art Museum, accession number 27.152) is a lost-wax cast that depicts the animals in a compact format, attached to a rectangular plaster base. The current composition of the Barye model in the J. Paul Getty Museum collection and later bronzes (including sand casts at the Walters Art Museum (accession number 27.4510) and Baltimore Museum of Art (object number 1996.46.45)) are significantly different from the *surtout* bronze, with changes to the shape, length, and posture of both animals and the addition of a larger rocky outcrop. Close examination under visible and ultraviolet light, comparisons of 3D surface scans, and radiographs led to the hypothesis that the Barye model was originally hollow and conformed to the more compact *surtout* composition, but was later broken into sections and reconfigured to be used as a working model for the elongated sand-cast versions. However, the radiographs of this highly complex object proved to be difficult to interpret with certainty, so definitive confirmation of this hypothesis was not possible.

The Barye model was scanned at the in-house facility of the J. Paul Getty Museum[31]. It was mounted on the rotation stage with the center of rotation positioned so that only a small part of the base would rotate out of the field of view on some radiographs. The part of the base not covered by all projections was not significant for the investigation of the hypotheses and therefore a lower image quality in that region was considered acceptable. Seventeen markers were inserted in foam and placed next to the object at the top and the bottom, to avoid overlap with the denser parts of the object in the radiographs (see Fig. 5b). Tube voltage and current were 450 kV and 2 mA, respectively. The exposure time was 2.85 seconds per capture, and to reduce noise, each radiograph was the result of averaging four captures. In total, 718 radiographs were recorded over two revolutions of the rotation stage.

The 3D CT reconstruction of the Barye model revealed several key features that could not have been observed with traditional radiography and other noninvasive examination techniques. One significant question at the start of the study was how closely the composition of the Barye model matched that of the *surtout* bronze. The radiographs showed small gaps that suggested that the rectangular base of the

original Barye model might have been embedded in added plaster to create the larger rocky outcrop. The CT reconstruction (see Fig. 5c) confirmed this observation quickly and easily; horizontal slices clearly reveal an embedded rectangular area of plaster of a different density than the surrounding plaster that matches the positioning of the original base, see Fig. 5d. Another initial inquiry was focused on the metal armature and how it was embedded in sections of plaster. 2D radiographs showed varying plaster densities in different areas, suggesting that the model may have originally been hollow and was later filled with a second pour of plaster; however, it was difficult to be certain that this was a valid interpretation because of overlapping features and digital artifacts in the radiographs. The 3D CT data, on the other hand, not only validated this theory by allowing clear observation of the two different plaster densities throughout the model but also revealed a third layer of plaster that was previously unidentified. The data allowed conservators to confirm that the object was initially a hollow sculpture with two layers of plaster: an initial fine, thin layer that was likely slushed into a mold to capture surface detail, and a secondary, thicker layer for support. This sculpture was then filled with a third layer of high porosity plaster in order to embed the metal armature after the reconfiguration (see Fig. 5e, red box, and 5f with an enlargement and arrows indicating the different layers). Large structural gaps that resulted from the reconfiguration but could not be easily replaced with plaster were instead filled with wax. This material change can be seen most prominently on the neck of the gnu, where the difference in density between the plaster and the wax is clear (see 5e, blue box). Modeling wax was also added to the surface of the gnu in several areas to alter the animal's musculature and match its reconfigured position.

Several other features that were difficult or impossible to observe in the 2D data were also discovered during the examination of the 3D reconstruction, such as the exact positioning of armature endpoints and density variation between materials in complicated internal regions. The case study of the Barye model proved to be successful, allowing Getty conservators to confirm aspects of the construction method for this object and to better document evidence of changes made by the artist.

## Discussion

Our major results are twofold. First, we developed a method that enables 3D CT scanning with standard 2D radiography equipment, significantly increasing the accessibility of CT imaging within the museum research field by making optimal use of available hardware. The novelty of our approach is that compared to existing CT methods, it does not rely on pre-calibrated system parameters and is flexible with respect to the hardware components. Second, the technique was used to perform CT imaging in the in-house X-ray suites in the J. Paul Getty Museum and the Rijksmuseum, to the best of our knowledge for the first time, without extra hardware investment. Until now, these systems had only been employed for radiography.

The interior of a cultural heritage object holds valuable information on the object's origin, artist's methods, previous conservation treatments and current state, which can be revealed by CT imaging without damaging the object. By deploying in-house X-ray systems, one can avoid the costly and difficult transportation of precious objects to CT facilities located in hospitals or laboratories. An important advantage of our method is that limited hardware investments are required, making it accessible to all museum research facilities with a standard radiography setup, for whom the purchase of dedicated CT systems is often out of reach. Our method incurs negligible costs and uses only the available basic X-ray equipment as well as small metal balls, foam, and tailored algorithms.

Inaccuracies in the geometrical parameters lead to blurring, shape distortion, and streaks in the resulting reconstruction image[32]. Therefore, several studies have investigated the calibration of existing CT systems. Marker-based approaches have been employed previously,

for example with motion correction in medical C-arm CT[33] and the geometrical calibration of CT systems[34–36]. In most cases, a dedicated calibration phantom is used, in which the position of the markers is precisely controlled during fabrication[37] or measured with high precision after fabrication[38,39]. Some approaches have more flexibility in the marker positions, for example when using an adaptable LEGO phantom[40] or arbitrary marker locations[41]. These methods usually rely on a pre-scan of a marker phantom. They moreover assume the rotation stage is sufficiently reliable to produce accurate equidistant rotation angles between radiographs and calculate the other system parameters[34]. This is an important difference with our method, which was designed to include the estimation of rotation angles such that there is no dependency on the accuracy of the hardware.

Calibration is also important for non-standard trajectories, which are for example encountered in robotic CT[42]. Highly flexible robotic arms have been designed that can allow for adapting the acquisition trajectory to the object. A calibration step is performed by tracking a reference object to compensate for inaccuracies in the trajectory[43]. Our method currently assumes a circular trajectory. In principle, it could be extended to include more degrees of freedom in the calculated parameters to facilitate the handling of more general acquisition trajectories.

Apart from marker-based methods, efforts have been made to compensate for inaccuracies in the acquisition parameters using optimization methods. These methods are usually applied to increase image quality by minor alterations in the parameters given by the CT system and are therefore dependent on the suitability of the internal features of the object (e.g., containing sharp edges)[44]. Other studies investigate methods to perform iterative reconstruction and alignment simultaneously[45]. The optimization is often applied to a subset of the parameters set used in our approach[36,45,46].

Obtaining a 3D reconstruction provides more information on the internal features than 2D radiographs. In the case of wood, for example, when a cross-section of sufficiently high resolution has been obtained, this could be used by dendrochronologists to measure the tree rings and date them through comparison with reference chronologies[47]. The resolution should be high enough to capture the thinnest rings in the transverse section of the sample[47,48].

Through the datasets acquired at three different museum research facilities and a micro-CT facility, we show the flexibility of our approach. An important feature of our method is that, in addition to the system parameters, the marker positions are included as parameters in this optimization. Therefore no tailored, specifically made calibration phantoms are needed and the marker foam can be adjusted to suit the object. The calculation of the angular increments makes this approach applicable to systems that cannot be relied upon to produce equidistant angles. If the system does produce equidistant angles, the markers could be used for a pre-calibration step to obtain the parameters that stay the same (source-detector-distance, object-detector-distance, detector tilts), eliminating the need for keeping markers in the scan with the object and using an inpainting step afterward (see Supplementary Figs. 8–10). Including the markers in the object scan, however, allows our approach to be applied in a wide range of uncalibrated X-ray setups and may also be used to validate assumptions on the system geometry of existing CT setups, e.g. by checking if the projection angles are indeed equally spaced.

The method presented in this manuscript has three main limiting factors. First, the resolution of the final image is dependent on the available hardware, mainly the X-ray tube focal spot size as well as the possible distances between the source, object, and detector, which will be different for each setup and scan. The current method provides a 3D reconstruction and does not determine an absolute scale. In order to perform measurements on the reconstruction, either one feature on the object, the distance between markers or an included dummy object with known size needs to be measured to adjust the absolute

scale. Second, the field of view of the detector determines the size of the objects that can be scanned. Third, the flexibility in motion of the hardware components will limit the possibility of performing tiled scans.

In future work, we would like to include horizontal and vertical tiling, which involves recording multiple datasets with different positions of the source and detector and computationally tying them together to image larger objects. Many factors can play a role when acquiring tiled scans. For systems where the detector can move independently from the source, stitching can be performed relatively easily. The projections are sampled based on the same source position and can therefore be stitched to on large projection[18]. A limiting factor here is the cone angle and fan angle of the beam, the detector may move to a position where it is not fully illuminated by the X-rays. For systems where the source and detector are linked (such as is the case in the BM and RM for instance) stitching becomes more complicated since the data is sampled based on different source positions and horizontal tiles have to be reconstructed simultaneously. This requires the data to be processed simultaneously by the reconstruction algorithm, for which stitching algorithms have been developed previously, for instance within the FleX-box toolbox[49] or Astra toolbox[50].

Notably, all instructions for the acquisition phase at the J. Paul Getty Museum took place in online meetings, without any need for the computer scientists to be physically present in the X-ray suite. The computational workflow was carried out afterward in Amsterdam. We aim to reduce the involvement of computer scientists in the processing workflow by further automating the method and providing a user-friendly interface, which would stimulate the adoption of our method in other research facilities. This would greatly increase the amount of knowledge gained from CT imaging in the cultural heritage sector in general and will also play an important role in bringing research results to public attention.

By expanding the capabilities of existing hardware with post-scan parameter derivation, CT imaging will become more accessible to a wider cultural heritage community, thus further bridging the gap between digital methods and cultural heritage research. Our method may also be useful for the development of portable X-ray systems since no pre-calibration of components is needed. Enabling CT scanning in pre-existing radiography setups in museums also increases the options for applying post-scan image processing methods to a wider range of objects.

The application of 3D X-ray imaging on a broader scale will challenge conservators and museum professionals to incorporate the previously inaccessible interiors of objects as part of their research on museum objects, encouraging new perspectives on how we investigate and conserve cultural heritage. Our method has shown the potential of computational methods to upgrade existing hardware with previously unimplemented capabilities. This step toward further integration of computational methods with traditional techniques will promote the development of both research fields.

## Methods
### Computed tomography reconstruction
X-ray imaging setups for scanning static objects typically consist of an X-ray source, an X-ray detector, and a rotation stage in between, on which the object is mounted. For our workflow, we use a cone-beam X-ray source and a digital flat-panel detector. The detector measures the intensity of the X-ray beam profile after it is attenuated by the object, resulting in a projection of the object's internal structure. Apart from the material composition and location of internal features within the object, the measurement of the projection image further depends on the geometrical system parameters: the precise location of the source, object, and detector, as well as the orientation of the object and the acquisition angles. A CT dataset consists of a set of these projection images, or *radiographs*, typically hundreds to thousands acquired

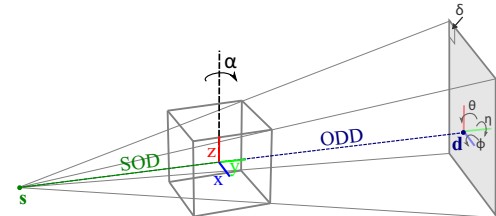

**Fig. 6 | Schematic representation of the X-ray setup.** X-ray setup with system components and parameters indicated: source (**s**), detector (**d**), source-detector-distance (SOD), object-detector-distance (ODD), coordinate system ($x, y, z$), rotation angle $\alpha$, detector pixel $\delta$ and detector tilts ($\eta, \theta, \phi$).

across a full rotational range. After data acquisition, a CT reconstruction algorithm computes a 3D volumetric image of the scanned object based on the acquired radiographs and the system parameters. In commercial CT systems, the system components are managed by high-quality computer-controlled motors to precisely control the orientation and timing of each radiograph. The rotation stage supporting the object is at an accurately known position and rotates at a constant speed around an axis that can be assumed to be exactly aligned with the vertical axis of the detector plane, enabling the use of the Feldkamp-David-Kress algorithm for efficient and accurate 3D reconstruction[51].

In contrast to dedicated CT scanners, when attempting to perform a CT scan using a basic X-ray imaging system designed for live radiography inspection, a range of parameters governing the geometry of the acquisition are unknown at the time of measurement (see Fig. 6). The 2D X-ray imaging system can be combined with any kind of rotation stage, with varying control mechanisms. This variety makes a solution that is independent of specific hardware components highly desirable.

The acquisition process can be modeled as a system of equations. The so-called *forward operator* $\mathbf{A}_\Theta$ contains all the geometric information on the scanning process and therefore depends on the vector $\Theta$, which contains the *unknown* system parameters, such as the distances between the source, center of rotation, and detector; the projection angles; and the detector tilts. The vector $\mathbf{x}$ is the digital representation of the object and $\mathbf{b}$ is the projection data acquired[52]. The goal is to find the representation of the image that leads to the acquired projection data, and in the process to minimize the difference between the forward projected image representation and the data:

$$\min_{\mathbf{x}} |\mathbf{A}_\Theta \mathbf{x} - \mathbf{b}|^2. \tag{1}$$

We first computationally derive the system parameters $\Theta$ using the marker-based approach detailed in the next section and then solve equation (1) by using the algebraic SIRT algorithm, a standard iterative reconstruction method in the CT field[53]. The SIRT algorithm operates by performing a gradient descent to minimize the residual, which is determined by forward-projecting the current estimate of the object representation and comparing it to the data.

### Marker-based post-scan parameter derivation
In our workflow, the markers are used not to refine given system parameters or calibrate an existing CT system, but rather to estimate all of the parameters necessary, thus obtaining 3D CT reconstructions from systems that were not designed for this purpose. In standard CT systems, the projection angles are equidistant. However, since we are working with non-calibrated systems, the projection angles can be non-equidistant and are therefore part of the parameter set that is derived. Moreover, the positions of the markers in the foam are configurable. They can therefore be easily adapted to the diversity of museum objects, and are also part of the parameter set. The positions

of the markers in the foam are not precisely controlled or measured but placed vertically spaced to avoid overlap on the radiographs. In the Supplementary Methods, we give some general guidelines on the positioning of the markers within the foam. On the radiographs, we can detect the projected marker location (PML). We aim to computationally find the positions of the markers and the system parameters to match these measured PMLs.

Given a set of system parameters and marker positions, we can calculate a *predicted PML*, by taking the intersection of a line through the source and the marker with the detector plane. The modeled system parameters are shown in Fig. 6 and given in Table 2. The positions of the markers in the foam are considered unknown and therefore constitute additional parameters. The aim is to find the system parameters and marker positions for which the predicted PML $\mathbf{p}_{ij}^{pred}(\Theta, \mathbf{m}_j) = (x_{ij}^{pred}(\Theta, \mathbf{m}_j), y_{ij}^{pred}(\Theta, \mathbf{m}_j))$ for each marker $j$ on radiograph $i$ is as close as possible to the measured PML $\mathbf{p}_{ij}^{meas} = (x_{ij}^{meas}, y_{ij}^{meas})$.

We therefore want to find the parameters $\Theta$ which minimize the following value:

$$\sum_i \sum_j \left| \mathbf{p}_{ij}^{meas} - \mathbf{p}_{ij}^{pred}(\Theta, \mathbf{m}_j) \right|^2 \qquad (2)$$

We approximate the parameters $\Theta$ by employing a least squares solver. These parameters are then used to obtain a 3D reconstruction of the object by using the algebraic SIRT algorithm to solve equation (1)[53]. Our method obtains a 3D reconstruction and does not determine the actual physical dimensions. Excluding voxel dimensions, this does not impact the CT reconstruction. An object of known size can be included in the acquisition or a feature on the object can be measured to determine the scale. Details regarding the theory underlying our method and the practical and computational implementation can be found in the Supplementary Methods.

**Radiography suites**

The setups that were used in this research (see Fig. 2) each consisted of an X-ray tube, rotation stage, and digital flat-panel detector. Below, we briefly describe the characteristics of each facility; the individual specifications can be found in Table 3.

- **British Museum, London** The British Museum X-radiography suite contains an Yxlon Access Y.100 industrial radiography system (Yxlon, Germany), with digital radiography and CT scanning capabilities. The system utilizes a Y.TU 450-D11 bipolar cone-beam X-ray tube, with a tungsten target, nominal tube voltage of 450 kV, and focal spot size of 0.4 mm at 700 W output. The X-rays are projected onto a PerkinElmer XRD 1621 AN15 ES flat-panel detector (40.96 × 40.96 cm), which consists of 2048 x 2048 pixels, 200 µm pixel pitch. The source and detector are suspended from a gantry by a retractable belt system, and they move together in the horizontal and vertical axes. The system provides feedback on the X-ray tube, detector, and turntable positions. The X-ray tube and detector positions are adjusted by the user with either a pendant or joystick system, and the turntable position is automatically controlled by the system software throughout CT acquisition. The turntable pauses during the acquisition of each radiograph, and the number of radiographs (thus the rotation angle per step) for a scan is predetermined by the user. CT reconstruction is conducted automatically by the system following a scan using the VGSTUDIO 3.2 software package (Volume Graphics, Germany).

- **J. Paul Getty Museum, Los Angeles** The system at the J. Paul Getty Museum is a radiography system for live inspection of objects. The X-ray source is a General Electric system, consisting of a pair of Isovolt Titan E generators driving a cone-beam bipolar Isovolt 450/10 X-ray tube, with a voltage range of 5-450 kV. There are two focal spot sizes of 5.5 mm and 2.5 mm, with a maximum power of 4.5 kW and 1.68 kW, respectively. The X-ray detector is a GE DXR250U-W digital panel with a detector area measuring 40.96 x 40.96 cm, with a pixel size of 200 µm yielding images of 2048 x 2048 pixels. The tube and detector are mounted on independent carriages on a remotely operable gantry. Images are acquired one at a time using GE Rhythm software. Objects are rotated on an Ortery Photocapture 360 M computer-controlled turntable with 1° rotation interval; 0.5° intervals are acquired by first acquiring 360 images at 1° interval, then manually rotating the turntable by approximately 0.5°, and then acquiring a second set of 360 images at 1° intervals. Coordination between the image capture software and the turntable control software is accomplished using RoboTask automation software.

- **Rijksmuseum, Amsterdam** The system is designed for live radiographic inspection of objects. The apparatus is a Balteau Baltograph X-ray system, which consists of a Baltograph Generator XSD225 with cone-beam X-ray tube TSD225/0, with a voltage range of 2−225 kV and a focal spot size of 1 mm (640 W max.) or 5.5 mm (3000 W max.), and Control Unit LS1. The X-rays are projected onto a flat-panel detector (Balteau Baltoscope FPDIGIT13-127), with a detector area of 19.5 cm x 24.4 cm, which consists of 1920 x 1536 pixels, with a pixel size of 127 µm. The

**Table 2 | Parameters used in the marker-based post-scan parameter derivation**

| | |
|---|---|
| Source location | $\mathbf{s} = (s_x, SOD, s_z)$ |
| Detector location | $\mathbf{d} = (d_x, d_y, d_z)$ |
| Detector tilt | $\theta, \phi$ |
| Detector in-plane rotation | $\eta$ |
| Detector pixel size | $\delta$ |
| Projection angles | $A = \{\alpha_0, ..., \alpha_{n-1}\}$ |
| Set of all system parameters | $\Theta = \{\mathbf{s}, \mathbf{d}, \theta, \phi, \eta, \delta, A\}$ |
| Marker positions | $\mathbf{m}_1, ..., \mathbf{m}_N$ |

**Table 3 | System specifications of the radiography suites**

| System characteristics | British Museum | J. Paul Getty Museum | Rijksmuseum | FleX-ray |
|---|---|---|---|---|
| Minimum focal spot size (mm) | 0.4 | 2.5 | 0.5 | 0.017 |
| Maximum voltage (kV) | 450 | 450 | 225 | 90 |
| Detector size (cm x cm) | 40.96 x 40.96 | 40.96 x 40.96 | 24.58 x 19.66 | 14.59 x 11.49 |
| Detector pixel size (µm) | 200 | 200 | 127 | 74.8 |
| Maximum source-detector distance (m) | 2.5 | 3 | 1.5 | 1.1 |
| Projection angles equidistant | Yes | No | No | Yes |
| Rotation mode | Start-stop | Start-stop | Continuous | Continuous |
| Approximate object size (cm x cm x cm) | 25 x 25 x 30 | 25 x 25 x 30 | 15 x 15 x 10 | 10 x 10 x 8 |

source and detector are mounted on either side of a gantry and are thus moved together. The system can record radiographs in mp4 video format, while the rotation stage moves continually. The angular increment per radiograph is not constant during the acquisition. There is moreover no way to determine accurately when a full 360˚ rotation has been recorded. The recording is continued long enough to make sure information is gathered over at least a full rotation. The motors are externally controlled by a Seifert DP435 system, using joysticks that control vertical and horizontal movement and tilts of the X-ray tube and detector, move the rotation stage in two directions, set the rotation speed, and control the rotational movement. For the last of these functions, the joystick needs to be manually pushed throughout the recording to make the stage move continually. There is, however, no feedback on the location of the components or displacement. It is not possible to accurately choose parameters such as rotation speed or locations of source and detector.

- **FleX-ray, Amsterdam** The cabinet-based system from TESCAN XRE is a highly flexible system designed to develop and test different acquisition trajectories. The system features a cone-beam microfocus X-ray point source with an energy range of 20–90 kV with a maximum of 90 W at 90 kV. The focal spot size is 17 μm. The flat-panel detector is a CMOS (complementary metal-oxide semiconductor) detector with CsI(Tl) scintillator (Dexela1512NDT), with 1944 × 1536 pixels (14.59 cm × 11.49 cm). The detector pixel size is 74.8 μm. Datasets can be recorded at angular intervals of 0.1˚ with continuous rotation.

## Data availability

The data of the wooden block generated in this study have been deposited in the *Zenodo* database: • The British Museum: https://doi.org/10.5281/zenodo.8379910 • The J. Paul Getty Museum: https://doi.org/10.5281/zenodo.8379880 • The Rijksmuseum: https://doi.org/10.5281/zenodo.8379870 • FleX-ray laboratory: https://doi.org/10.5281/zenodo.10557034 The data of the case study *Python killing a Gnu* generated in this study have been deposited in the *Zenodo* database: https://doi.org/10.5281/zenodo.8379913.

## Code availability

Our code is publicly available on GitHub at https://github.com/fgbossema/markers, in the form of a python package installable with pip, and an archived version at *Zenodo*: https://doi.org/10.5281/zenodo.8379920.

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

## Acknowledgements

We are grateful to our Rijksmuseum colleagues who provided feedback during the initial design process: Paul van Duin, Isabelle Garachon, Jolanda van Iperen, and Robert van Langh. We thank Allard Hendriksen (CWI) and Jan-Willem Buurlage (CWI) for fruitful discussions in the first stages of this project. We thank Maria Sherwood-Smith (Leiden University) for proofreading sections of our manuscript. We thank David Barick for proofreading and language editing services. We thank Rosa de Boer (University of Utrecht) and Enny van Beest (University College London) for providing feedback from the perspective of scientists from different research fields. We thank Katrien Keune (Rijksmuseum, University of Amsterdam) for providing useful comments and insights on a draft of the manuscript. We are grateful to TESCAN XRE for their collaboration regarding the FleX-ray laboratory. This research is part of the Impact4Art project, which is supported by the Netherlands Institute for Conservation, Art and Science (NICAS) and the Dutch Research Council (NWO) (project number 628.007.033, K.J.B.). F.G.B. received support for a research position at the British Museum in London from the Prins Bernhard Culture Foundation, the Jo Kolk Study Foundation, the Catherine van Tussenbroek Foundation, and the European Women in Mathematics Association (EWM). F.G.B. is currently supported by the Michelien Gerritzen Fund/Rijksmuseum Fund. F.G.B. is also a fellow at the Netherlands eScience Center, Amsterdam.

## Author contributions

F.G.B., K.J.B., W.J.P., and T.v.L. conceived the approach and experiments. F.G.B. developed the main code, R.v.L. contributed marker locating and inpainting code. F.G.B. and J. Dorscheid performed data acquisition at the Rijksmuseum facility. F.G.B. and D.O'F. performed data acquisition at the British Museum facility. A.H. performed data acquisition at the J. Paul Getty Museum facility. F.G.B. analyzed the data and performed 3D reconstructions of all datasets. M.C. analyzed the 3D reconstructions of the case study object and drafted the section with information and results about the case study object. J. Dorscheid performed species identification of the wooden block. F.G.B. drafted the main manuscript and appendices. K.J.B., W.J.P., and T.v.L. reviewed and edited the manuscript. All authors reviewed and commented on the manuscript.

## Competing interests

The authors declare no competing interests.
