## [Peer Review File · Nature Communications]

Enabling 3D CT-scanning of cultural heritage objects using only in-house 2D X-ray equipment in museumsRESPONSE TO REVIEWERS' COMMENTS

We thank the reviewers for their constructive feedback, which has assisted us in improving this article. Below we will address the comments by reviewers one by one. Reviewer comments copied in black, our replies in blue. In the main text, the changes are indicated in blue.

Reviewer #1 (Remarks to the Author):

A new method is presented that extends the capabilities of standard X-ray inspection systems commonly available in museums and galleries. The principle of the method is that the object of interest is imaged from a number of views such that a 3D reconstruction of the object can be performed to obtain a visualization of the internal structure similar to that of expensive CT systems.

I can definitely recommend this article for publication after minor revision. The proposed inspection protocol is similar to robotic CT where the planned trajectory is corrected/calculated based on the tracking of the reference object. Therefore some relevant publication should be referred (for instance

<https://link.springer.com/article/10.1007/s10921-022-00888-9> or <https://iopscience.iop.org/article/10.1088/2631-8695/ac8224>)

We thank reviewer 1 for their compliments and suggestions. Thank you for drawing our attention to robotic CT, we have included this topic in the section on 'related methods' in the Discussion (p.10 l.258).

In terms of the range and accuracy of scanning trajectories, standard radiographic systems do not have the capabilities of robotic systems. Therefore, the proposed procedure for detecting the position of the X-ray tube relative to the detector is quite adequate. At the same time, it would be useful to give at least an approximate accuracy of the calculated marker positions.

Thank you for the suggestion. We have included simulation experiments to investigate the accuracy and precision of the found marker positions in the revised manuscript. Below we describe the simulation experiments and present the results, which we have also included in the revised manuscript in the Supplementary Material, section C.1.

We simulated projected marker locations (PMLs) by forward projecting 3D marker locations. To simulate an incorrectly found center of the PML, we added gaussian noise with mean 0 and standard deviation from 0 up to 5 pixels to the PML. Next, we ran our parameter retrieval optimisation to obtain estimated 3D marker locations. Since our method does not assume the projection angles and other system parameters are known, the found 3D marker locations may have a slightly different orientation, vertical position, and scaling compared to the original marker positions. These variations do not affect the reconstruction quality, so to be able to measure the quality of the found marker locations, we compensate for them before comparing the found marker locations with the original marker locations used for the forward projection. For each choice of standard deviation, we ran this simulation with ten different random seeds, which influences the noise added to the PMLs, to obtain Figure 1 showing standard deviation (in pixels) versus the average error in the calculated marker positions (in mm). The average error is lower than the voxel

size (0.13mm) except for a few outliers. In the practical datasets included in the manuscript, we found that the PML identification can be trusted to locate the centers within this error range.

Figure 1: **Results of the simulation experiment.** Boxplot showing average errors in the calculated marker positions (y-axis) when adding gaussian noise to the PML (standard deviation of noise on the x-axis). The boxplot shows the median (blue line), interquartile range (lightblue box), which shows where 50% of the data points around the median fall, and minima and maxima of the data (black horizontal bars).

It would also be useful to indicate how large objects and what shapes can be inspected by the proposed procedure/used X-ray inspection system.

We have included this information in Table 3.

As for the obtained spatial resolution, it should correspond to the size of the emission spot of the X-ray tube used. The actual resolution of the tomographic reconstruction could probably be estimated from the size of the smallest bubbles in the plaster in Fig. 4.

We agree that the size of the emission spot of the X-ray tube is an important factor in the resolution. Other influencing factors are for instance the distances between the source, object and detector. We have included this in the Discussion (p.11, l.290).

Reviewer #2 (Remarks to the Author):

Authors introduce a novel workflow for acquiring 3D CT images of cultural heritage objects using basic X-ray equipment which is often available in museums. This is in contrast to advanced and fully calibrated CT systems which are normally available only in specialized laboratories. The authors propose a complete workflow for computing 3D CT from a series of 2D radiography measurements. In their novel approach they surround the object of interest with a piece of foam and they place markers in this piece. The markers are small metal balls. They then acquire several 2D radiograph from different angle using a rotation stage. Due to the simplicity of the set-up, the projections are uncalibrated, therefore, they use advanced computational techniques based on the estimation of the position of the markers to estimate the angles of the projections and then they can obtain the complete 3D CT scan. This means that they get 3D information of the interior of the object. They applied their technique on different cultural heritage objects and with three different museums. I find the case study of the “Python Killing a GNU” particularly interesting and compelling. In particular they show that their approach allowed to show that gaps in the reconfigured neck of the gnu were filled with wax rather than plaster.

The work is clearly novel in its “system engineering” aspect in that authors put together an end-to-end system and clearly show how hardware complexity can be reduced by modifying the acquisition process (e.g., by adding markers) and by increasing the complexity of the computational methods involved. They combine this with compelling case studies that fully validate the effectiveness of their approach.

We thank reviewer 2 for their feedback and compliments.

My only recommendation is that they expand a bit more on the possible limitations of their current approach. They can do that in the supplement or in the discussion. In particular, the objects they have scanned so far are relatively small. They mention that through mosaicking they can potentially scan bigger objects but mosaicking is sometimes difficult. So it would be helpful to know what is a realistic maximum size of an object that they can scan with the current approach.

Thank you for these suggestions. We have included an estimate of the realistic object sizes that can be scanned with the systems used for this work in Table 3. We have moreover extended the discussion with a subsection on the limitations of our approach, including comments on the challenges of mosaicking (p.11, l.290 & p11, l.302).

Their discussion on number and location of markers is also a bit vague at the moment. It would be helpful if they could expand a bit more on that. In particular is there a minimum distance recommended for markers? That in turn could give an indication of the right number of markers for objects' size.

For the scans in this article, we have used a minimum distance between markers of 1cm. Using live radiographic inspection, we checked that there were as few projections with overlapping markers as possible. This facilitates the labeling and tracking of the markers. We have added a step-by step description of this approach to the Supplementary Material, since this may be of value for readers who want to implement this method (SM, p. 5, l.93).

Minor

Line 56 replace “limited to select cases” with “limited to selected cases”

Line 107 replace “refer to section 4” with “refer to Section 4”

We thank the reviewer for their attention to detail and have implemented these suggestions.

Reviewer #3 (Remarks to the Author):

Dear Francien Bossema et al

This paper presents a novel method for generating 3D X-ray CT of museum objects, using conventional X-ray equipment in three museums, but with the addition of small metal markers around the object being scanned. The metal markers anchor the position of the many X-ray images taken at numerous angles of rotation, and through developing mathematical models for the 3D reconstruction, the authors successfully build 3D models of the internal structure of a museum object, but at a much more affordable cost. This is a highly novel and innovative approach, for inspecting the internal structure of the sculpture that is examined in this way. It is well written, clear, and thorough.

We thank the reviewer for their feedback and suggestions.

However, in the first experiments, comparing the output of three X-ray installations in England, Netherlands and USA, respectively, using a “small wooden block”, the authors suggest that this new CT scanning method would be suitable for dendrochronological analysis. It is stated: “The tree rings and the saw cut are visible in all reconstructions”.

I would like to see a statement of what genus the wooden block is, how many rings are observed in the object, have they been measured by a dendrochronologist, and have the virtual cross-sections similarly been measured, and the results compared? What is the average growth rate of the rings of the block, how narrow is its narrowest ring? While this is not necessarily the main point of the article and the experiments carried out, it should not be claimed that the method is suitable for dendrochronology if this has not yet been demonstrated. In the discipline of dendrochronology very narrow tree rings are frequently encountered, so it requires particularly good imagery so that these narrow rings can be detected in a CT virtual cross-section (see e.g. Bill et al 2012). Would it be possible for a statement of how narrow the rings are on the wooden block, and how this new CT technique might be applicable for non-invasive dendrochronological studies.

Bill, J., Dalen, K.S., Daly, A. & Johnsen, Ø. Dendro CT—dendrochronology without damage. *Dendrochronologia* 30, 223–230 (2012)

We agree with the reviewer that the experimental results for the wooden block included in our manuscript are not suitable for demonstrating dendrochronology capabilities. We now see that

our paper was too strongly worded on this matter, since we have not measured the wood. We consider a full dendrochronology experiment to be out-of-scope for the current article, as also pointed out by the reviewer.

Instead, in the revised manuscript we have added a discussion on what would be needed to make the step from acquiring CT images using our approach to an actual dendrochronological measurement (p10, l.270). Moreover, we have included the genus of the wooden block (*Taxus* ssp.) (p.5, l.119).

Also, in your Data statement, I have not been able to access the records in Zenodo.

That is correct. We propose to publish the data upon acceptance of the article, but have already reserved DOI's for the respective datasets and included them in our paper. For reviewing purposes we uploaded all data to a folder. The link to these is included in a statement right after our data and code availability statement: "The code and data will be published upon acceptance of the article. For reviewing purposes, all data and code can be found here: <https://surfdive.surf.nl/files/index.php/s/CWHws9YvfputsFo>." (p.17, l.519). In the final version we will remove this link and make the Zenodo links accessible. We apologize if this was unclear.

Reviewer #4 (Remarks to the Author):

The authors present a method to include CT capabilities at the available radiography setups at three museums, one of which is actually also a CT instrument. The latter illustrates that the objective of this manuscript is not that challenging. It is hard to assess to what extent their approach gives better results than a more straightforward approach, as such results are not shown (and as far as I can read such study is not performed). The absence of such thorough investigation and comparison, along with the quite straightforward concept presented here, are my main concerns, from which I conclude the manuscript is not sufficiently scientifically sound and novel to justify publication in Nature Communications. Nonetheless, I feel the method is worth reporting, but the structure of the manuscript (notably its dependency on case studies) makes it more suited for journals like Scientific Reports. Hereafter I will list a few comments (in chronological order), both minor and major, to enable the authors to improve the work for the revised, transferred or resubmitted version

We thank the reviewer for their suggestions for the improvement of our article.

- The names of the museums are not an added value in the title.

The method we present in this article is especially well suited for implementation within the museum context, which is characterized by (i) challenges in object mobility (in particular taking objects outside the museum) and (ii) availability of varied and versatile X-ray imaging hardware, as well as various mechanical stages and turntables. Through mentioning the well-known museums that collaborated in this study in the title, we aim to emphasize that these considerations apply even in large museums with considerable resources. Although we would prefer to keep the title as it is, we have provided the editor with a shortened alternative: 'Enabling 3D CT-scanning of cultural heritage objects using only basic in-house 2D X-ray equipment in the museum'.

- line 46: state why a CT scan needs a dedicated system. Furthermore, this claim should be supported by the results as well, by showing the failure of a non-dedicated system, as mentioned earlier. However, I believe that even a "blunt" approach of applying FDK on these datasets with rough estimations of the parameters (distances as measured by a ruler, angles as positioned by eye and with a laser cross, ...) would already yield decent results, particularly for the GM system considering its huge spot size

A CT scan needs a dedicated system because the reconstruction depends on the acquired radiographs as well as several geometrical parameters that are part of the system output in well-calibrated systems. We have added this to the introduction.

We agree with the reviewer that there are multiple possible approaches to obtain reconstructions with a non-dedicated system. The choice for the approach shown in our article was motivated by the wish to get a flexible acquisition method, putting most of the work into the post-processing phase.

We thank the reviewer for the suggestion to visually show the effect of the parameters on the reconstruction. We think this is interesting for the reader and have therefore included the images presented below in the Supplementary Material, section C.2.

Below we show the effect of a 'blunt' approach on the reconstructed image. We applied the FDK algorithm to the data of the wooden block acquired at the J. Paul Getty museum (GM). Within the FDK algorithm, the data is assumed to have been acquired over a circular trajectory with equidistant projection angles. In the case of the GM data, the projection angles are not equidistant. Therefore, assuming that the angles are equidistant gives errors in the angular increment, which leads to wrongly backprojected radiographs as can be seen in the FDK reconstruction (left). As shown, the FDK method is surprisingly sensitive to the errors in the projection angles. Our method estimates the angular increments and therefore a much better reconstruction is obtained (right).

Two types of errors that occur in the museum systems covered in our article are errors in the rotation speed and missing projection angles. Below we show the effect of these errors on a dataset of the wooden block recorded at a micro-CT facility, the FleX-ray laboratory, located at the Center for Mathematics and Computer Science in Amsterdam.

In the image below we show how an error in the rotation speed affects the FDK reconstruction. In this case, the rotation stage is rotating slower than a full rotation during the acquisition. From left to right is shown a reconstruction when the stage reaches a rotation of 357.5, 355, 325.5 and 350 degrees during an acquisition. In the reconstruction we see that this leads to wrongly back projected angles and the slower the rotation stage, the worse the reconstruction.

The image below shows the reconstruction when a few consequent projection angles missing, from left to right 2, 5, 10 and 20 missing angles around 60 degrees. This shows the effect of missing angles in one place only on the reconstruction. In the datasets of the museums, angles can be missing in multiple sections.

- in line 60, the authors claim that the systems are focused on a specific object dimensions range. This is only given by the size of the detector, which is also the case for the setups mentioned here. In fact, the detectors used here are also commonly used in commercial μ CT systems, hence no added value is given here. I was hoping that the manuscript would also tackle this problem by integrating a stitching algorithm, which I believe would be possible.

The possible object dimension is in most systems indeed currently limited by the detector size, because this determines the field of view. The other limiting factors are the space within the scanner and the distance between source and detector, since the object needs to fully rotate

between them. The space within a (micro)CT-cabinet is more constrained than in the X-ray imaging facilities in the museums. We have clarified this in the Introduction (p.2 I.61)

Many factors can play a role when doing mosaicking of scans. For systems where the detector can move independently from the source, stitching can be performed relatively easily. The projections are sampled from the same cone and can therefore be stitched to on large projection. (see figure 1 below). A limiting factor there is the cone angle and fan angle of the beam, the detector may move to a position where it is not fully illuminated by the X-rays.

For systems where the source and detector are linked (such as is the case in the BM and RM for instance) stitching becomes more complicated, since the data is sampled from different cones and horizontal tiles have to be reconstructed simultaneously.

Stitching algorithms have been developed previously, for instance within the FleX-box toolbox [Kostenko et al, 2020] or Astra toolbox [Van Aarle et al, 2016]. To fully tackle stitching within the framework of radiography systems that were not built for CT requires a fair number of complex steps and is therefore outside the scope of the current manuscript. We have extended the discussion with the considerations mentioned here (p.11, I.302).

Figure 1. Tiled scans by moving the detector allow for stitching of the projections to one large projection. Image adapted from [Bossema et al, 2021].

References

Kostenko, A., Palenstijn, W.J., Coban, S.B., Hendriksen, A., van Liere, R. & Batenburg, K.J., Prototyping X-ray tomographic reconstruction pipelines with FleXbox. *SoftwareX* 11, 100364 (2020)

Van Aarle, W., Palenstijn, W.J., Cant, J., Janssens, E., Bleichrodt, F., Dabrovolski, A., De Beenhouwer, J., Batenburg, K.J., Sijbers, J., Fast and flexible X-ray tomography using the ASTRA toolbox. *Opt. Express* **24**, 25129–25147 (2016)

Bossema, F.G., Coban, S.B., Kostenko, A., Van Duin, P., Dorscheid, J., Garachon, I., Hermens, E. Van Liere, R., Batenburg, K.J., Integrating expert feedback on the spot in a time-efficient explorative CT scanning workflow for cultural heritage objects, *Journal of Cultural Heritage*, Vol. 49, p38-47 (2021)

- In the next paragraph (starting line 66) there is some repetition about the flexibility
We have reformulated this sentence to avoid repetition (p.2, l.70).

- The results are compared with the result from - essentially - one of the 3 systems under investigation. This is a strange reasoning, and a thorough comparison with another system (a dedicated μ CT system) would be beneficial

We have collected a dataset of the wooden block at a micro-CT facility, the FleX-ray laboratory, located at the Center for Mathematics and Computer Science in Amsterdam. We have updated the images in the article to include the data from this scan.

- The effects of the inpainting are not investigated. A comparison between the results with inpainting and without the markers (from a conventional μ CT system as mentioned earlier) is in my opinion essential, preferably in a worst-case slice

We agree with the reviewer that it is of interest to the reader to see the effect of the inpainting. The datasets we recorded contain markers, we therefore show the reconstruction with the original radiographs and the inpainted radiographs. We compare the cross-sections from a reconstruction of the FleX-ray dataset with and without inpainting in the following image:

We also compare the cross-sections from a reconstruction of the dataset recorded at the J. Paul Getty Museum with and without inpainting in the following image:

These images have been added to the Supplementary Material, section C.3. When using our methods, it is easy to perform both a reconstruction with the original radiographs and the inpainted radiographs. Therefore the user can choose which reconstruction serves them best, since the effect of inpainting can differ per object and placement of the markers.

- In line 128 the authors mention the angular increment. In the supplementary material, the axes are not labeled, so it is hard to interpret. However, if the non-linearity is rather small, the effect on the resolution is minimal. Furthermore, this issue could have its origin merely in the implementation or the parameters of the motors.

Thank you for pointing this out, the axis labels have been added. Indeed, when the non-linearity is small, the effect is small. However, the systems at the GM and the RM produce irregular angular intervals due to the interplay between the motor operation and the sampling and storage frequency characteristics of the detection system. The added value of our method is that despite these errors (which do occur in actual deployed X-ray systems), a CT reconstruction can be obtained by retrieving estimated angles directly from the acquired projection data.

- The authors claim in line 135-136 that it is hard to pinpoint the exact reasons. However, when starting from a near-perfect reconstruction (from a μ CT system), it is perfectly possible to simulate each of the imperfections separately and pinpoint their influence. As mentioned, my feeling is that the focal spot size will be dominating. The efficiency of the parameter estimation could also be investigated by doing the analysis on a commercial system which should be perfectly aligned, by starting from a wrong initial guess for example. Furthermore, it is definitely technically feasible to bring your own rotation stage and align it perfectly, which would already compensate for one aspect of the setup.

We agree with the reviewer that spot size is an important factor. In regular well-calibrated CT systems spot size is indeed the most influencing factor and has become almost synonymous with resolution. Inaccuracies in the geometrical parameters however, also lead to blurring, shape distortion and streaks in the resulting reconstruction image [Abella et al., 2021]. Depending on the accuracy of the hardware, either the spot size or calibration parameters become dominating. In the systems under investigation, the unknown alignment is a dominant factor as shown by the comparison to a straightforward FDK above. In the revised manuscript, we have included a number of additional experiments in the Supplementary Material. First, we included simulations to investigate the robustness of our method (SM, section C.1). Second, we include reconstructions from the FleX-ray dataset with varying degrees of misalignment in the angles (SM, section C.2). The FDK reconstruction algorithm is quite sensitive to these errors. Therefore, an accurate estimate of the angles is of importance. If there is the possibility to bring in a reliable rotation stage, this could be a way to avoid having to calculate the angles. However, it may be a challenge to integrate new hardware with existing setups. A strong point of our method is that no new hardware is needed.

References

Abella, M., Martinez, C., Garcia, I., Moreno, P., De Molina, C. & Desco, M. Tolerance to geometrical inaccuracies in CBCT systems: A comprehensive study. *Medical Physics*, **48**, 6007–6019 (2021).

- Lines 142 and onwards bring me to my main concern: would this result be impossible to achieve without the markers? It would be very interesting to see the reconstructions of the reconstruction of the dataset without markers, both very straightforward and with some effort in optimizing the geometrical parameters. Making the datasets available to other researchers or suppliers of reconstruction software (e.g. MITOS) to have an independent reconstruction would be one option.

We agree with the reviewer that there are other possible approaches to this problem. We have extended the section on related methods with references to post-scan parameter optimization (p.10 l.262). These methods often rely on the suitability of the object's internal features to determine the reconstruction quality (e.g. sharp edges) or determine a subset of the parameters estimated by our method. Our marker-based approach is designed to be robust, generic and straightforward, to make it applicable in a wide range of X-ray facilities. Upon acceptance of the article all acquired datasets will be made available open access on Zenodo (See Data and Code availability statement) and we would be happy for other researchers to try different methods on the data.

- Figure 3 is very difficult to assess. The line profile helps a bit, though only confirms the rough analysis that the GM reconstruction is very smooth. Numerical conclusions, zoomed-in slices or any other means to estimate the real quality (both smoothing and other reconstruction artefacts) are key in publishing this kind of work

We have added zoomed-in slices to the figure with the reconstruction results (p.7 figure 4). We have moreover added simulation experiments to the Supplementary Material to investigate the robustness of our method (SM. section C.1).

- Section 2.3 reports a "case study". Though this is interesting to read, I find it a big portion of the manuscript for a high-impact journal and the added value is limited. In my opinion, this should rather be supplementary information.

The purpose of section 2.3 is to specifically demonstrate the added value of obtaining 3D CT images for answering relevant research questions in (technical) art history. We point out that before the implementation of our method, this information was not accessible to the J. Paul Getty museum. Since Nature Communications is an interdisciplinary journal that is read by scientists from many backgrounds, we moreover feel that the case study will appeal to the readership.

- In line 177 the authors mention the object rotates out of the field of view. This brings me to an earlier comment concerning the flexibility and limitation to certain sizes, hence this statement is a contradiction with an earlier statement in the manuscript.

The reviewer is right. In order to get a reconstruction of the entire object, it should stay in the field of view. The data collected for parts that rotate out of the view can lead to truncation artifacts in the reconstruction. In the case study, the part that rotates out of the field of view is a small part of the base, which was not needed to assess the questions about the case study. We have clarified this in the section (p.6 l.187).

- in line 252, the authors mention pre-calibration. How difficult and time-consuming would it be to do a similar pre-calibration of the setup (using only a limited number of projections)? I assume that in a real-world application of these systems for CT, the geometry won't be changed every scan, so the overhead would probably be minimal, while not having the issue of inpainting. This method of course doesn't allow for a retrieval of the angular increment, but this can also be exploited as it allows to investigate the effect of this specific misalignment

In the case of cultural heritage objects, it is often necessary to adjust the acquisition geometry to the object. Due to the wide variety of objects, it can change with every new object. If the system produces reliable (equidistant) angular increments, a pre-calibration step is certainly a possibility. The main issue with the systems that are encountered in museum setups is that the angular intervals are unknown. In these cases, a pre-calibration step could only provide the other parameters and a different post-processing step would be needed to obtain an estimate of the angular intervals. We have included these considerations in the Discussion (p.11 l.280).

- in line 259, the authors reveal that this work is not so much a novelty, but rather an implementation. I believe that with limited effort, the implementation of a basic CT modality at these systems would also allow for CT data acquisition at very similar quality levels (as I believe the main issue for the quality is not the alignment of the geometry, which is in fact very accurate (see Table 3 in SI)

In our previous replies, we have shown that depending on the system characteristics, calibration can be a limiting factor. This is especially so for irregular angular intervals. The key strength of our approach is that it will work regardless of the specific system.

- As an iterative reconstruction method is already used, essentially minimizing the reconstruction error, do you even need markers? Isn't it possible to include small uncertainties on the geometry in the reconstruction algorithm?

We agree that there are other approaches possible and we have extended the review of other possible methods with markerless methods in the Discussion (p.10 l.258). Our choice for this approach is motivated by the application to the museum field, with flexibility and generalizability in mind, as explained more extensively in previous replies.

- The authors mention calibration phantoms and their design requirements, but these are usually used to determine very accurately the voxel size as well. To determine only the misalignments, such requirements are not imposed

We have extended the review of other possible methods in the Discussion (p.10, l.245-268).

- line 365: is this a direct detection? Looking at the spec sheet, it uses a Gadox screen?

Thank you for pointing out this error, we have removed 'direct'.

- line 379: please mention the type and vendor (looking at the pixel size, a Varian / Varex system)

We have added the information (p.15 l.451).

REVIEWER COMMENTS

Reviewer #1 (Remarks to the Author):

A new method is presented that extends the capabilities of standard X-ray inspection systems commonly available in museums and galleries. The principle of the method is that the object of interest is imaged from a number of views such that a 3D reconstruction of the object can be performed to obtain a visualization of the internal structure similar to that of expensive CT systems.

I can definitely recommend this article for publication after minor revision. The proposed inspection protocol is similar to robotic CT where the planned trajectory is corrected/calculated based on the tracking of the reference object. Therefore some relevant publication should be referred (for instance <https://link.springer.com/article/10.1007/s10921-022-00888-9> or <https://iopscience.iop.org/article/10.1088/2631-8695/ac8224>)

In terms of the range and accuracy of scanning trajectories, standard radiographic systems do not have the capabilities of robotic systems. Therefore, the proposed procedure for detecting the position of the X-ray tube relative to the detector is quite adequate. At the same time, it would be useful to give at least an approximate accuracy of the calculated marker positions. It would also be useful to indicate how large objects and what shapes can be inspected by the proposed procedure/used X-ray inspection system. As for the obtained spatial resolution, it should correspond to the size of the emission spot of the X-ray tube used. The actual resolution of the tomographic reconstruction could probably be estimated from the size of the smallest bubbles in the plaster in Fig. 4.

Reviewer #2 (Remarks to the Author):

Authors introduce a novel workflow for acquiring 3D CT images of cultural heritage objects using basic X-ray equipment which is often available in museums. This is in contrast to advanced and fully calibrated CT systems which are normally available only in specialized laboratories. The authors propose a complete workflow for computing 3D CT from a series of 2D radiography measurements. In their novel approach they surround the object of interest with a piece of foam and they place markers in this piece. The markers are small metal balls. They then acquire several 2D radiograph from different angle using a rotation stage. Due to the simplicity of the set-up, the projections are uncalibrated, therefore, they use advanced computational techniques based on the estimation of the position of the markers to estimate the angles of the projections and then they can obtain the complete 3D CT scan. This means that they get 3D information of the interior of the object. They applied their technique on different cultural heritage objects and with three different museums. I find the case study of the "Python Killing a GNU" particularly interesting and compelling. In particular they show that their approach allowed to show that gaps in the reconfigured neck of the gnu were filled with wax rather than plaster.

The work is clearly novel in its "system engineering" aspect in that authors put together an end-to-end system and clearly show how hardware complexity can be reduced by modifying the acquisition process (e.g., by adding markers) and by increasing the complexity of the computational methods involved. They combine this with compelling case studies that fully validate the effectiveness of their approach.

My only recommendation is that they expand a bit more on the possible limitations of their current approach. They can do that in the supplement or in the discussion. In particular, the objects they have scanned so far are relatively small. They mention that through mosaicking they can potentially scan bigger objects but mosaicking is sometimes difficult. So it would be helpful to know what is a realistic maximum size of an object that they can scan with the current approach. Their discussion on number and location of markers is also a bit vague at the moment. It would be helpful if they could expand a bit more on that. In particular is there a minimum distance recommended for markers? That in turn could give an indication of the right number of markers for objects' size.

Minor

Line 56 replace "limited to select cases" with "limited to selected cases"

Line 107 replace "refer to section 4" with "refer to Section 4"

Reviewer #3 (Remarks to the Author):

Dear Francien Bossema et al

This paper presents a novel method for generating 3D X-ray CT of museum objects, using conventional X-ray equipment in three museums, but with the addition of small metal markers around the object being scanned. The metal markers anchor the position of the many X-ray images taken at numerous angles of rotation, and through developing mathematical models for the 3D reconstruction, the authors successfully build 3D models of the internal structure of a museum object, but at a much more affordable cost. This is a highly novel and innovative approach, for inspecting the internal structure of the sculpture that is examined in this way. It is well written, clear, and thorough.

However, in the first experiments, comparing the output of three X-ray installations in England, Netherlands and USA, respectively, using a "small wooden block", the authors suggest that this new CT scanning method would be suitable for dendrochronological analysis. It is stated: "The tree rings and the saw cut are visible in all reconstructions".

I would like to see a statement of what genus the wooden block is, how many rings are observed in the object, have they been measured by a dendrochronologist, and have the virtual cross-sections similarly been measured, and the results compared? What is the average growth rate of the rings of the block, how narrow is its narrowest ring? While this is not necessarily the main point of the article and the experiments carried out, it should not be claimed that the method is suitable for dendrochronology if this has not yet been demonstrated. In the discipline of dendrochronology very narrow tree rings are frequently encountered, so it requires particularly good imagery so that these narrow rings can be detected in a CT virtual cross-section (see e.g. Bill et al 2012). Would it be possible for a statement of how narrow the rings are on the wooden block, and how this new CT technique might be applicable for non-invasive dendrochronological studies. Also, in your Data statement, I have not been able to access the records in Zenodo.

Bill, J., Dalen, K.S., Daly, A. & Johnsen, Ø. Dendro CT—dendrochronology without damage. *Dendrochronologia* 30, 223–230 (2012)

review by Aoife Daly

Reviewer #4 (Remarks to the Author):

The authors present a method to include CT capabilities at the available radiography setups at three museums, one of which is actually also a CT instrument. The latter illustrates that the objective of this manuscript is not that challenging. It is hard to assess to what extent their approach gives better results than a more straightforward approach, as such results are not shown (and as far as I can read such study is not performed). The absence of such thorough investigation and comparison, along with the quite straightforward concept presented here, are my main concerns, from which I conclude the manuscript is not sufficiently scientifically sound and novel to justify publication in *Nature Communications*. Nonetheless, I feel the method is worth reporting, but the structure of the manuscript (notably its dependency on case studies) makes it more suited for journals like *Scientific Reports*.

Hereafter I will list a few comments (in chronological order), both minor and major, to enable the authors to improve the work for the revised, transferred or resubmitted version

- The names of the museums are not an added value in the title.

- line 46: state why a CT scan needs a dedicated system. Furthermore, this claim should be supported by the results as well, by showing the failure of a non-dedicated system, as mentioned

earlier. However, I believe that even a "blunt" approach of applying FDK on these datasets with rough estimations of the parameters (distances as measured by a ruler, angles as positioned by eye and with a laser cross, ...) would already yield decent results, particularly for the GM system considering its huge spot size

- in line 60, the authors claim that the systems are focused on a specific object dimensions range. This is only given by the size of the detector, which is also the case for the setups mentioned here. In fact, the detectors used here are also commonly used in commercial μ CT systems, hence no added value is given here. I was hoping that the manuscript would also tackle this problem by integrating a stitching algorithm, which I believe would be possible.

- In the next paragraph (starting line 66) there is some repetition about the flexibility

- The results are compared with the result from - essentially - one of the 3 systems under investigation. This is a strange reasoning, and a thorough comparison with another system (a dedicated μ CT system) would be beneficial

- The effects of the inpainting are not investigated. A comparison between the results with inpainting and without the markers (from a conventional μ CT system as mentioned earlier) is in my opinion essential, preferably in a worst-case slice

- In line 128 the authors mention the angular increment. In the supplementary material, the axes are not labeled, so it is hard to interpret. However, if the non-linearity is rather small, the effect on the resolution is minimal. Furthermore, this issue could have its origin merely in the implementation or the parameters of the motors.

- The authors claim in line 135-136 that it is hard to pinpoint the exact reasons. However, when starting from a near-perfect reconstruction (from a μ CT system), it is perfectly possible to simulate each of the imperfections separately and pinpoint their influence. As mentioned, my feeling is that the focal spot size will be dominating. The efficiency of the parameter estimation could also be investigated by doing the analysis on a commercial system which should be perfectly aligned, by starting from a wrong initial guess for example. Furthermore, it is definitely technically feasible to bring your own rotation stage and align it perfectly, which would already compensate for one aspect of the setup.

- Lines 142 and onwards bring me to my main concern: would this result be impossible to achieve without the markers? It would be very interesting to see the reconstructions of the reconstruction of the dataset without markers, both very straightforward and with some effort in optimizing the geometrical parameters. Making the datasets available to other researchers or suppliers of reconstruction software (e.g. MITOS) to have an independent reconstruction would be one option.

- Figure 3 is very difficult to assess. The line profile helps a bit, though only confirms the rough analysis that the GM reconstruction is very smooth. Numerical conclusions, zoomed-in slices or any other means to estimate the real quality (both smoothing and other reconstruction artefacts) are key in publishing this kind of work

- Section 2.3 reports a "case study". Though this is interesting to read, I find it a big portion of the manuscript for a high-impact journal and the added value is limited. In my opinion, this should rather be supplementary information.

- In line 177 the authors mention the object rotates out of the field of view. This brings me to an earlier comment concerning the flexibility and limitation to certain sizes, hence this statement is a contradiction with an earlier statement in the manuscript.

- in line 252, the authors mention pre-calibration. How difficult and time-consuming would it be to do a similar pre-calibration of the setup (using only a limited number of projections)? I assume that in a real-world application of these systems for CT, the geometry won't be changed every scan, so the overhead would probably be minimal, while not having the issue of inpainting. This method of course doesn't allow for a retrieval of the angular increment, but this can also be exploited as it allows to investigate the effect of this specific misalignment

- in line 259, the authors reveal that this work is not so much a novelty, but rather an implementation. I believe that with limited effort, the implementation of a basic CT modality at these systems would also allow for CT data acquisition at very similar quality levels (as I believe the main issue for the quality is not the alignment of the geometry, which is in fact very accurate (see Table 3 in SI)

- As an iterative reconstruction method is already used, essentially minimizing the reconstruction error, do you even need markers? Isn't it possible to include small uncertainties on the geometry in the reconstruction algorithm?

- The authors mention calibration phantoms and their design requirements, but these are usually used to determine very accurately the voxel size as well. To determine only the misalignments,

such requirements are not imposed

- line 365: is this a direct detection? Looking at the spec sheet, it uses a Gadox screen?
- line 379: please mention the type and vendor (looking at the pixel size, a Varian / Varex system)

REVIEWERS' COMMENTS

Reviewer #1 (Remarks to the Author):

The paper has been significantly improved in accordance with the reviewers' comments. The results are very nice. I very much appreciate that the method has been tested on multiple devices, so that the capabilities and limitations of the proposed method are well documented. Also, it is very helpful that the authors provided the python code without restrictions. I highly recommend the paper for publication.

Only some minor problems remained:

Figure 3. subfigure d) looks flipped compared to the other subfigures.

I recommend

Line 54 Maybe „stages“ instead of „motors“.

Line 110 What is mean by the sentence „and removing the markers by inpainting“? Interpolating of data inside of the labeled region?

Line 144 Regarding authors statement: "As we were working with uncalibrated systems in which multiple hardware and software factors may play a role, we cannot exactly pinpoint the reasons for the differences in image quality." Calibration of the system geometry itself is not reason for different image quality. It is quite clear correlation between tube spot size and spatial resolution in the paper "Note that X-ray tube manufacturers are using different definition of the spot size".

Line 385: I recomend "In standard CT systems, the projection angles are equidistant." instead of "In standard CT systems, the projection angles are required to be equidistant." It is because it is not required in many cases.

Data collection was done in a rather standard way, i.e. with a stable position of the rotary table and detector. "Only" the uncertainty of the axis and rotation angle is addressed in the presented method. On the other hand, the CT measurement could be performed with a more general CT scan path and the proposed method should work (without rotary stage for instance). It should be mentioned in discussion.

Reviewer #2 (Remarks to the Author):

I was already happy with the paper and my minor criticisms have now been addressed. Si I support publication of this paper.

Reviewer #3 (Remarks to the Author):

Dear Authors

Thank you for your well-argued responses to my comments.

With best wishes

Aoife

Reviewer #4 (Remarks to the Author):

I would like to thank the authors for their extensive replies to the reviews, including mine. Before I start, I'd like to mention that I did not read the manuscript in full again nor did I thoroughly check the actual implementations of some of the answers, as I rely on the authors' goodwill to improve the manuscript.

I have to say that I am still not convinced about the fundamental novelty of the method rather than a (well-performed) development, and as a result whether it is suited for NCOMMS.

Apart from that, I have the following remarks concerning the replies of the authors:

- The title is in my opinion much better without mentioning the museum names. I understand mentioning well-known museums draws the attention, this is in my opinion something for popular science, not for a scientific journal

- The missing angle artefact seems to be identical to the last-angle-artefact. Were these missing angles ignored, as inpainting empty frames could solve this issue quite well. This obviously implies knowledge about the number of angles, but this is often feasible by timestamping the image acquisition.

- I thank the authors for including a discussion about stitching, which I acknowledge is too much to tackle in detail, but in my opinion needed mentioning as is done now

- I believe the investigation of the effect of the inpainting is not completely fair, as it is also the effect of the marker itself that needs to be investigated, and therefore should be compared with the sample without a marker. Looking at the figure "Flex-ray inpainted", I still notice some artefacts (comparing with the 357.5° figure above), which may be misleading when interpreting the data. This is mentioned by the authors in the rebuttal ("Therefore the user can choose which reconstruction serves them best, since the effect of inpainting can differ per object and placement of the markers"), but I don't find this statement in the main manuscript.

RESPONSE TO REVIEWERS' COMMENTS

Below we will address the comments by reviewers one by one. Reviewer comments copied in black, our replies in blue.

Reviewer #1 (Remarks to the Author):

The paper has been significantly improved in accordance with the reviewers' comments. The results are very nice. I very much appreciate that the method has been tested on multiple devices, so that the capabilities and limitations of the proposed method are well documented. Also, it is very helpful that the authors provided the python code without restrictions. I highly recommend the paper for publication.

Only some minor problems remained:

Figure 3. subfigure d) looks flipped compared to the other subfigures.

Thank you for your attention to detail. In this dataset the foam pieces with markers were inadvertently placed the other way round. This is why the wooden block is in the same position, but the markers are in flipped positions. This does not matter for the post-processing pipeline.

I recommend

Line 54 Maybe „stages“ instead of „motors“.

Thank you, we have incorporated this suggestion.

Line 110 What is mean by the sentence „and removing the markers by inpainting“? Interpolating of data inside of the labeled region?

Yes, that is correct. More information is given in the Supplementary Information, Section A.2.2.

Line 144 Regarding authors statement: “As we were working with uncalibrated systems in which multiple hardware and software factors may play a role, we cannot exactly pinpoint the reasons for the differences in image quality.” Calibration of the system geometry itself is not reason for different image quality. It is quite clear correlation between tube spot size and spatial resolution in the paper "Note that X-ray tube manufacturers are using different definition of the spot size".

We agree with the reviewer that there is a strong correlation between the tube spot size and spatial resolution. In the case of the J. Paul Getty Museum dataset this effect is clearly visible, therefore we explicitly mention this in the manuscript (p5, l134). Blurring in the reconstructed image can also be caused by inaccuracies in the geometrical parameters [Abella et al., 2021], which we demonstrate in the Supplementary Information (Supplementary figures 6 and 7). To distinguish between the effects caused by the different influencing factors, extensive experiments would be required that, in our opinion, fall outside the scope of the current manuscript.

References

Abella, M., Martinez, C., Garcia, I., Moreno, P., De Molina, C. & Desco, M. Tolerance to geometrical inaccuracies in CBCT systems: A comprehensive study. *Medical Physics*, **48**, 6007–6019 (2021).

Line 385: I recommend "In standard CT systems, the projection angles are equidistant." instead of "In standard CT systems, the projection angles are required to be equidistant." It is because it is not required in many cases.

We have removed 'are required' as suggested.

Data collection was done in a rather standard way, i.e. with a stable position of the rotary table and detector. "Only" the uncertainty of the axis and rotation angle is addressed in the presented method. On the other hand, the CT measurement could be performed with a more general CT scan path and the proposed method should work (without rotary stage for instance). It should be mentioned in discussion.

We thank the reviewer for the suggestion and have incorporated a statement in the Discussion to this effect: 'Our method currently assumes a circular trajectory. In principle, it could be extended to include more degrees of freedom in the calculated parameters to facilitate handling of more general acquisition trajectories.'

Reviewer #2 (Remarks to the Author):

I was already happy with the paper and my minor criticisms have now been addressed. So I support publication of this paper.

Reviewer #3 (Remarks to the Author):

Dear Authors
Thank you for your well-argued responses to my comments.
With best wishes
Aoife

Reviewer #4 (Remarks to the Author):

I would like to thank the authors for their extensive replies to the reviews, including mine. Before I start, I'd like to mention that I did not read the manuscript in full again nor did I thoroughly check the actual implementations of some of the answers, as I rely on the authors' goodwill to improve the manuscript.

I have to say that I am still not convinced about the fundamental novelty of the method rather than a (well-performed) development, and as a result whether it is suited for NCOMMS.

Apart from that, I have the following remarks concerning the replies of the authors:

- The title is in my opinion much better without mentioning the museum names. I understand mentioning well-known museums draws the attention, this is in my opinion something for popular science, not for a scientific journal

Thank you, we have edited the title accordingly.

- The missing angle artefact seems to be identical to the last-angle-artefact. Were these missing angles ignored, as inpainting empty frames could solve this issue quite well. This obviously implies knowledge about the number of angles, but this is often feasible by timestamping the image acquisition.

We thank the reviewer for the attention to detail. Supplementary figure 8 is indeed showing a very similar effect to figure 7 and is therefore superfluous. We have removed Supplementary figure 8. The goal of figure 6 and 7 is to show the type of artefacts that can be encountered when an imperfect rotation stage is used for CT imaging. We have clarified this further in the Supplementary Information (Section C.2).

- I thank the authors for including a discussion about stitching, which I acknowledge is too much to tackle in detail, but in my opinion needed mentioning as is done now

Thank you. We agree.

- I believe the investigation of the effect of the inpainting is not completely fair, as it is also the effect of the marker itself that needs to be investigated, and therefore should be compared with the sample without a marker. Looking at the figure "Flex-ray inpainted", I still notice some artefacts (comparing with the 357.5° figure above), which may be misleading when interpreting the data. This is mentioned by the authors in the rebuttal ("Therefore the user can choose which reconstruction serves them best, since the effect of inpainting can differ per object and placement of the markers"), but I don't find this statement in the main manuscript.

We agree that it would be interesting to investigate the effect of the inpainting with a dataset that was recorded without the markers. The main goal of this work is to obtain a 3D CT scan with 2D radiography equipment and therefore we feel that experiment is out-of-scope for the current manuscript. We have included the note mentioned by the reviewer in the section about inpainting in the Supplementary Information (Section A.2.2.).